# CYCLIP: Cyclic Contrastive Language-Image Pretraining

**Shashank Goel**[*]
UCLA
shashankgoel@ucla.edu

**Hritik Bansal**[*]
UCLA
hbansal@ucla.edu

**Sumit Bhatia**
MDSR Lab, Adobe Systems
sumit.bhatia@adobe.com

**Ryan A. Rossi**
Adobe Research
ryrossi@adobe.com

**Vishwa Vinay**
Adobe Research
vinay@adobe.com

**Aditya Grover**
UCLA
adityag@cs.ucla.edu

## Abstract

Recent advances in contrastive representation learning over paired image-text data have led to models such as CLIP [44] that achieve state-of-the-art performance for zero-shot classification and distributional robustness. Such models typically require joint reasoning in the image and text representation spaces for downstream inference tasks. Contrary to prior beliefs, we demonstrate that the image and text representations learned via a standard contrastive objective are not interchangeable and can lead to inconsistent downstream predictions. To mitigate this issue, we formalize *consistency* and propose CYCLIP, a framework for contrastive representation learning that explicitly optimizes for the learned representations to be *geometrically consistent* in the image and text space. In particular, we show that consistent representations can be learned by explicitly symmetrizing (a) the similarity between the two mismatched image-text pairs (cross-modal consistency); and (b) the similarity between the image-image pair and the text-text pair (in-modal consistency). Empirically, we show that the improved consistency in CYCLIP translates to significant gains over CLIP, with gains ranging from $10\% - 24\%$ for zero-shot classification accuracy on standard benchmarks (CIFAR-10, CIFAR-100, ImageNet1K) and $10\% - 27\%$ for robustness to various natural distribution shifts. The code is available at https://github.com/goel-shashank/CyCLIP.

## 1 Introduction

The ability to learn general-purpose representations from diverse data modalities is a long-standing goal of artificial intelligence (AI) [4, 32]. In this regard, recent instantiations such as CLIP [44], ALIGN [29], and BASIC [41] have scaled up vision-language contrastive pretraining to jointly learn image and text embeddings, by exploiting an enormous amount of paired image-text data on the web. Post pretraining, these embeddings exhibit impressive zero-shot classification performance [13] and robustness to natural distribution shifts [48, 57, 24, 26]. Recently, these embeddings have been extended to text-guided generation of natural images [47, 12, 38, 46] and transferred to modalities such as 3-D shapes [50] by emphasizing the interchangeability of the image and text embeddings.

In the context of vision-language pretraining, the standard contrastive learning objective aims to maximize the similarity between matched image-text pairs ("positives") against all the mismatched image-text pairs ("negatives") [45, 7, 40, 22]. While such an objective aligns the true image-text pairs, it poses no constraints on the overall geometry of all data pairs, including the mismatched

36th Conference on Neural Information Processing Systems (NeurIPS 2022).

---

[*]Equal Contribution

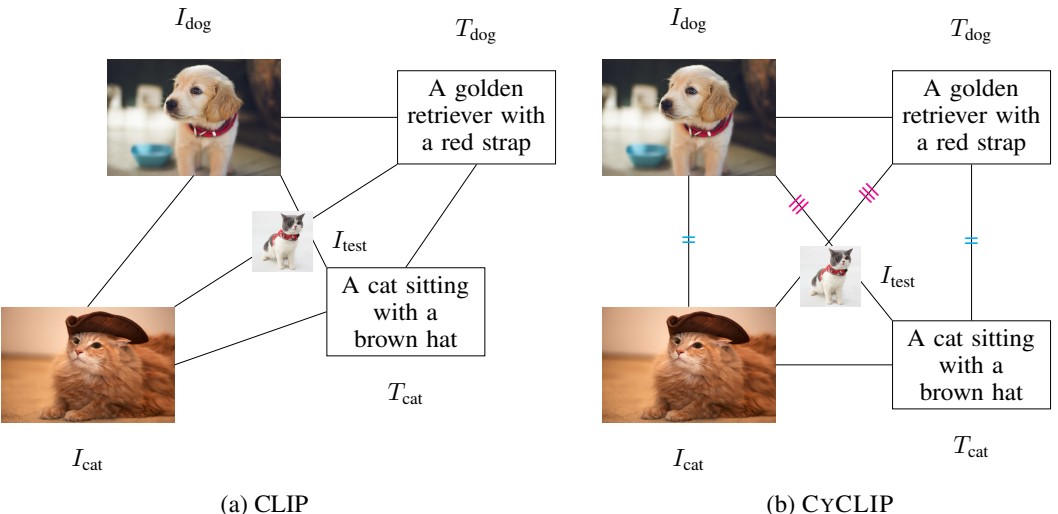

Figure 1: An illustration of the planar geometry of the learned representations of image-text pairs by (a) CLIP and (b) CYCLIP. The edges indicate the distance between the representations i.e., $d(e_1, e_2) = 1 - \langle e_1, e_2 \rangle$, where $\langle \cdot, \cdot \rangle$ is the inner product. CYCLIP is cyclic consistent between image-text pairs as the in-modal distances, $d(T_{\text{cat}}, T_{\text{dog}}) \sim d(I_{\text{cat}}, I_{\text{dog}})$, and the cross-modal distances, $d(T_{\text{cat}}, I_{\text{dog}}) \sim d(I_{\text{cat}}, T_{\text{dog}})$, are similar to each other unlike CLIP. Due to explicit consistency constraints, the test image of a cat is classified as a cat in the image as well as the text space.

pairs and pairs within the same modality. In Figure 1 (a), we illustrate this effect where matched image-text pairs, $(I_{\text{dog}}, T_{\text{dog}})$ and $(I_{\text{cat}}, T_{\text{cat}})$, get close to each other but the overall geometry of pairwise distances can be highly irregular (see e.g., $(I_{\text{dog}}, T_{\text{cat}})$ and $(I_{\text{cat}}, T_{\text{dog}})$). If we use such representations for downstream inference, such irregularities can translate into inconsistent reasoning in the image and text spaces. For example, CLIP designs proxy captions for class labels and uses the most similar class caption to perform zero-shot classification for images; using the default captions in Figure 1 (a), this would imply that a test image $I_{\text{test}}$ gets classified as a dog in the image space even when a simple nearest neighbor classifier in the text space would correctly infer the label to be a cat.

To mitigate these challenges, we propose **Cy**clic **C**ontrastive **L**anguage-**I**mage **P**retraining (CYCLIP), a framework that imposes additional geometric structure on the learned representations. Specifically, given two image-text pairs, we augment the contrastive learning objective with two symmetrization terms. The first term provides for in-modal consistency by encouraging the distance between the two image embeddings to be close to the distance between the corresponding text embeddings. The second term for the cross-modal consistency that encourages the distance between the image and text embedding from the first and second pairs respectively to be close to the distance between the text and image embeddings from the first and second pairs respectively. As shown in Figure 1 (b), if representations of any two image-text pairs, $(I_{\text{dog}}, T_{\text{dog}})$ and $(I_{\text{cat}}, T_{\text{cat}})$ exactly satisfy both forms of cyclic consistency, then we can guarantee that any test image $I_{\text{test}}$ respects the ordering of distances in both image and text spaces (i.e., if $d(I_{\text{test}}, I_{\text{dog}}) > d(I_{\text{test}}, I_{\text{cat}})$, then $d(I_{\text{test}}, T_{\text{dog}}) > d(I_{test}, T_{\text{cat}})$).

Empirically, we demonstrate that the improved consistency in CYCLIP translates to improvements over CLIP. In all cases, we pre-train our models on the Conceptual Captions 3M dataset[52]. On zero-shot classification, we observe that CYCLIP improves over CLIP by 10.2% on ImageNet1K, 10.6% on CIFAR-10 and 23.9% on CIFAR-100 respectively. Further, CYCLIP outperforms CLIP with an average relative gain of $+17\%$ on ImageNet natural distribution shift benchmarks. We further analyze the improved performance of CYCLIP and find that the additional geometric structure in the representation space better captures the coarse and fine-grained concept hierarchies of datasets.

Our contributions are as follows:

1. We analyze contrastive learning for representation learning jointly over image and text modalities. We identify a critical shortcoming in the geometry of the learned representation space that can lead to inconsistent predictions in image and text domains.

2. We propose CYCLIP, a simple and effective framework for contrastive representation learning with two additional cycle consistency constraints for mitigating the above issue.

3. We demonstrate that CYCLIP achieves significant empirical improvements over CLIP on zero-shot classification and robustness benchmarks. We further explain these improvements by analyzing the impact of consistency on the hierarchical structure of datasets.

## 2 Cycle Consistent Representation Learning

### 2.1 Preliminaries

We are interested in using text supervision to learn general-purpose visual representations that can be generalized to downstream predictive tasks. To this end, there have been several recent advances in language-image pretraining concerning model architectures, training objectives, and sources of supervision. Our work is most closely related to Contrastive Language-Image Pretraining (CLIP) [44] which combines many such advances in a highly scalable and generalizable learning framework.

CLIP is trained on millions of images with their captions scraped from the web. Formally, we consider a dataset $S \subset \mathcal{I} \times \mathcal{T}$ consisting of pairs $(I_j, T_j)$ where $I_j$ is a raw image and $T_j$ is a text caption. We use $\mathcal{I}$ and $\mathcal{T}$ to denote the domain of images and text, respectively. The CLIP architecture consists of 3 components: (i) an image encoder network, $f_I : \mathcal{I} \mapsto \mathbb{R}^d$, to encode the raw image into an embedding vector of dimension $d$, (ii) a text encoder network, $f_T : \mathcal{T} \mapsto \mathbb{R}^d$, to encode the raw text into an embedding vector of dimension $d$, (iii) a contrastive objective that pulls the embeddings of paired image-caption pairs together while pushing apart embeddings of unmatched pairs.

Formally, during training, consider a batch of $N$ image-captions pairs, $\{I_j, T_j\}_{j=1}^N$, where $I_j$ and $T_j$ represent the raw image and text pair, respectively. The image embedding $I_j^e \in \mathbb{R}^d$ and text embedding $T_j^e \in \mathbb{R}^d$ are obtained by passing $I_j$ and $T_j$ through the image encoder $f_I$ and text encoder $f_T$, respectively; i.e. $I_j^e = f_I(I_j)$ and $T_j^e = f_T(T_j)$. Further, we assume they are normalized to have unit $\ell_2$-norm. The contrastive objective in CLIP aims to align the image and text representations by minimizing the loss function $\mathcal{L}_{\text{CLIP}}$ shown below:

$$
\mathcal{L}_{\text{CLIP}} = -\frac{1}{2N} \sum_{j=1}^N \log \underbrace{\left[ \frac{\exp\left(\langle I_j^e, T_j^e \rangle / \tau\right)}{\sum_{k=1}^N \exp\left(\langle I_j^e, T_k^e \rangle / \tau\right)} \right]}_{\text{Contrasting images with the texts}} - \frac{1}{2N} \sum_{k=1}^N \log \underbrace{\left[ \frac{\exp\left(\langle I_k^e, T_k^e \rangle / \tau\right)}{\sum_{j=1}^N \exp\left(\langle I_j^e, T_k^e \rangle / \tau\right)} \right]}_{\text{Contrasting texts with the images}} \quad (1)
$$

where $\langle \cdot, \cdot \rangle$ represents the inner product, and $\tau$ is a trainable temperature parameter. CLIP and its variants can be used to perform zero-shot image classification, i.e., classifying test images into categories not seen at training time. We first transform each category into a suitable caption (e.g., the airplane category in CIFAR-10 can be expressed as 'a photo of an airplane'). Then, the similarity of the test image to each caption is computed (e.g., cosine distance), and the model predicts the category for which the image-caption similarity is the highest.

### 2.2 Inconsistent Representation Learning in CLIP

As illustrated in Figure 1 (a), the standard contrastive objective in CLIP can learn image-text representations such that the predicted labels for the test image are different in the image and text spaces. Here, we reason about such inconsistencies more formally in the context of downstream classification. As discussed above, we can predict a label in the text embedding space (zero-shot setting) by selecting the label that is closest to the test image ($P_T$). Additionally, for classification in the image embedding space, if we had access to a labeled training set, then one natural way to infer the predicted label ($P_I^k$) of a test image $I_{test}$ is by taking a majority vote from the true labels associated with the k-nearest training images. Formally, we define a consistency score that measures the synchrony between the predicted labels in the image and text spaces as:

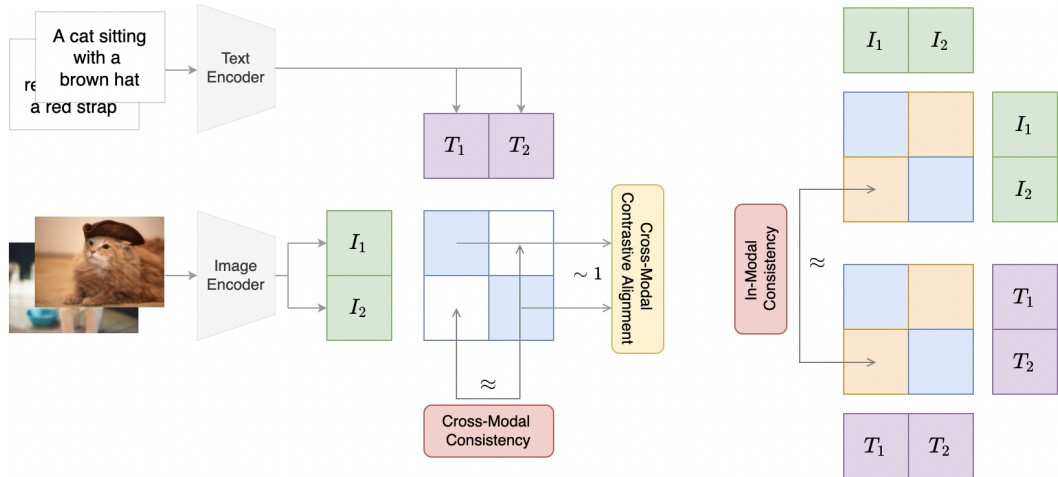

Figure 2: Illustrative overview for CYCLIP ($N = 2$). It consists of 3 major components: (a) cross-modal contrastive alignment, (b) cross-modal consistency, and (c) in-modal consistency. Only (a) is present in CLIP, whereas our proposed regularizers in (b) and (c) mitigate inconsistency.

$$\text{Consistency Score}_k = \frac{1}{N} \sum_{j=1}^{N} \mathbb{1} \left[ P_I^k(I_j) = P_T(I_j) \right] \tag{2}$$

where N is the number of test images. In our experiments (discussed in detail in §3), we found the CLIP's consistency score ($k = 1$) to be 44%, 16%, and 16% on the standard benchmarks CIFAR-10, CIFAR-100, and ImageNet1K, respectively, showing a very high degree of disagreement in the image and text spaces. In the following section, we describe our approach to alleviate the inconsistent inference problem and quantitatively show that our solution improves the consistency score in §4.1.

## 2.3 Cycle Consistent Representation Learning via CYCLIP

We showed that the visual representations learned by CLIP could be inconsistent when used for inference in the image and text spaces. To mitigate this problem, we propose CYCLIP, a learning framework that builds upon CLIP by augmenting the contrastive loss in Eq. 1 with additional geometric consistency regularizers. The intuition follows directly from Figure 1 (b), where we showed that inconsistency in the image and text spaces could be eliminated if we symmetrize the similarity between the two mismatched image-text pairs and the similarity between the image-image pair and the text-text pair. We formalize this intuition with two consistency regularizers.

(1) The **cross-modal consistency** regularizer reduces the gap in the similarity scores between the embeddings of all the mismatched image-text pairs in a batch, two at a time:

$$\mathcal{L}_{\text{C-Cyclic}} = \frac{1}{N} \sum_{j=1}^{N} \sum_{k=1}^{N} \left( \langle I_j^e, T_k^e \rangle - \langle I_k^e, T_j^e \rangle \right)^2. \tag{3}$$

(2) The **in-modal consistency** regularizer reduces the gap in the similarity scores between the embeddings of all combinations of image pairs and their corresponding text pairs in a batch:

$$\mathcal{L}_{\text{I-Cyclic}} = \frac{1}{N} \sum_{j=1}^{N} \sum_{k=1}^{N} \left( \langle I_j^e, I_k^e \rangle - \langle T_k^e, T_j^e \rangle \right)^2. \tag{4}$$

Hence, our overall loss for CYCLIP is given as:

$$\mathcal{L}_{\text{CYCLIP}} = \mathcal{L}_{\text{CLIP}} + \lambda_1 \mathcal{L}_{\text{I-Cyclic}} + \lambda_2 \mathcal{L}_{\text{C-Cyclic}} \tag{5}$$

where $\lambda_1 > 0$ and $\lambda_2 > 0$ are hyperparameters controlling the importance of the in-modal and cross-modal cyclic consistency regularizers relative to the contrastive loss in CLIP. We can also

characterize the effect of the regularizers in terms of symmetrizing the in-modal and cross-modal similarity matrices, as illustrated in Figure 2. Note that the optimal solution to the contrastive loss formulation would push the similarity between the normalized embeddings of the matched pairs towards 1 while forcing all other pairs of similarities to 0, thereby also symmetrizing the cross-modal similarity matrix and minimizing the cross-modal consistency loss. However, this idealized scenario does not occur in practice, and we find that explicit regularization via cycle-consistency in CYCLIP facilitates improved learning, as we show in our experiments.

## 3 Experiments

**Setup**: We use Conceptual Captions 3M [52] (CC3M) image-caption pairs as the source of multimodal pretraining data for all our models. Note while this dataset is smaller than the custom dataset (400 million pairs) used in the original work on CLIP [44], it is suitable for our available data and compute and has been used for benchmark evaluations in many subsequent works on language-image pretraining [5, 33, 37, 56]. Following prior work [44], our CLIP models use ResNet-50 as the image encoder and a transformer architecture as the text encoder. Further, we train our models from scratch for 64 epochs on 4 V100 GPUs with a batch size of 128 and an initial learning rate of 0.0005 with cosine scheduling and 10000 warmup steps. The dimension of the image and text embeddings is 1024. For CYCLIP, we use $\lambda_1 = 0.25$ and $\lambda_2 = 0.25$ across all our experiments.

### 3.1 Zero-Shot Transfer

We compare the zero-shot performance of CLIP and CYCLIP on standard image classification datasets: CIFAR-10, CIFAR-100 [31], and ImageNet1K [49]. We follow the evaluation strategy suggested by [44] for zero-shot classification using prompt engineering. For each dataset, we use the names of the classes to form a set of natural sentences such as 'a photo of the {class name}', 'a sketch of the {class name}' and more. These are passed through the text encoder to get a set of text embeddings for that class. This set of text embeddings are $\ell_2$-normalized, averaged, and further $\ell_2$-normalized to obtain a single text embedding for that class. For a given image, the image embedding is obtained as described in §2. The class whose text embedding (as described above) is closest to the test image is taken to be the predicted label. The zero-shot performance of the models is presented in Table 1.

Table 1: Zero-shot TopK classification accuracy (%) where K$\in \{1, 3, 5\}$

|  | CIFAR-10 | | | CIFAR-100 | | | ImageNet1K | | |
| --- | --- | --- | --- | --- | --- | --- | --- | --- | --- |
|  | Top1 | Top3 | Top5 | Top1 | Top3 | Top5 | Top1 | Top3 | Top5 |
| CLIP | 46.54 | 78.22 | 91.16 | 18.69 | 34.72 | 43.97 | 20.03 | 33.04 | 39.35 |
| CYCLIP | **51.45** | **79.57** | **91.80** | **23.15** | **41.46** | **50.66** | **22.08** | **35.98** | **42.30** |
| %GAIN | +10.6 | +1.7 | +0.7 | +23.9 | +19.4 | +15.2 | +10.2 | +8.9 | +7.5 |

We observe that the CYCLIP outperforms CLIP across all the datasets and on all TopK metrics, with gains in the range of 10% - 24% for K= 1. Our results on zero-shot transfer indicate the usefulness of having geometrical consistency for improved downstream performance of CLIP.

### 3.2 Robustness to Natural Distribution Shifts

One of the major successes of CLIP was its state-of-the-art performance on the natural distribution shift benchmarks. These benchmarks include images depicting sketches, cartoons, adversaries generated using attacks on trained ImageNet models. In Table 2, we evaluate the zero-shot classification accuracy of CYCLIP on four natural distribution shift benchmarks for the ImageNet dataset: ImageNetV2 [48], ImageNetSketch [57], ImageNet-A [27], and ImageNet-R [25].

For most of the distribution shift benchmarks, both CLIP and CYCLIP undergo a significant reduction in their zero-shot performance compared to the original ImageNet1K dataset (last three columns in Table 1). However, we observe that CYCLIP outperforms CLIP on all of the datasets considered in

Table 2: Zeroshot Classification on Natural Distribution Shifts (%)

| | ImageNetV2 | | | ImageNetSketch | | | ImageNet-A | | | ImageNet-R | | |
|---|---|---|---|---|---|---|---|---|---|---|---|---|
| | Top1 | Top3 | Top5 | Top1 | Top3 | Top5 | Top1 | Top3 | Top5 | Top1 | Top3 | Top5 |
| CLIP | 16.91 | 29.28 | 34.99 | 10.37 | 19.15 | 24.20 | 4.23 | 11.35 | 16.88 | 24.32 | 39.69 | 47.20 |
| CYCLIP | **19.22** | **32.29** | **38.41** | **12.26** | **22.56** | **28.17** | **5.35** | **13.53** | **19.51** | **26.79** | **42.31** | **50.03** |
| %GAIN | +13.7 | +10.3 | +9.8 | +18.2 | +17.8 | +16.4 | +26.5 | +19.2 | +15.6 | +10.2 | +6.6 | +6.0 |

this experiment by a significant margin of improvement (10 - 27%). This result indicates that having cyclic consistency in the learned representations preserves the robustness on the traditional datasets.

## 3.3 Linear Probing

While the primary focus of CLIP and CYCLIP is zero-shot generalization, we can also assess if the benefits of our cyclic consistency constraints in mitigating inconsistency can be recovered with extra in-domain and in-modality supervision i.e., in the presence of in-distribution training samples from in-domain visual datasets. To this end, we conduct an additional experiments on linear probing where we fit a linear classifier on the representations learned by the visual encoder (ResNet-50) of CLIP and CYCLIP on a range of image classification datasets.

Table 3: Transfer CLIP and CYCLIP to 14 downstream visual datasets using linear probing. Our CYCLIP performs marginally better on 9 out of 14 datasets. For training ImageNet1K, we use a random subset of 50K images from its original training dataset.

| | Caltech101 | CIFAR-10 | CIFAR-100 | DTD | Aircraft | Flowers102 | Food101 | GTSRB | ImageNet1K | OxfordPets | SST2 | StanfordCars | STL10 | SVHN | Average |
|---|---|---|---|---|---|---|---|---|---|---|---|---|---|---|---|
| CLIP | 79.80 | **78.26** | 54.85 | 59.02 | **28.00** | **83.50** | 54.44 | 69.72 | 35.93 | **57.66** | **53.82** | 20.00 | 89.23 | 47.28 | 57.96 |
| CYCLIP | **80.33** | 76.98 | **55.74** | **63.44** | 27.86 | 82.96 | **54.96** | **71.70** | **37.12** | 56.82 | 53.74 | **22.14** | **90.10** | **48.01** | **58.71** |

We present our results in Table 3. We find that both CLIP and CYCLIP can recover most of the performance lost due to inconsistency when provided extra in-domain and in-modality supervision, with CYCLIP marginally outperforming the CLIP on 9 out of 14 visual datasets.

## 4 Analysis

Previously, we demonstrated the gains of CYCLIP over CLIP on downstream tasks that involve joint reasoning over the image and text spaces. In the current section, we wish to better understand the relative behavior of the two models on a set of challenging tasks.

### 4.1 Consistency in Image and Text Spaces

We begin by quantitatively measuring the inconsistency problem illustrated in Figure 1. That is, we wish to evaluate to what extent are the predictions in the image-text space (zero-shot) consistent with the ones made purely within the image space, as measured by our consistency metric in Eq. 2.

Table 4 presents our results over standard benchmarks (CIFAR-10, CIFAR-100, ImageNet1K). The consistency score is calculated over 10K, 10K, and 50K testing images of the CIFAR-10, CIFAR-100 and ImageNet dataset respectively. We use 50K samples from the training set of each dataset for k-Nearest Neighbor prediction. CYCLIP is more consistent than CLIP across all the datasets as we explicitly symmetrize the cross-modal and in-modal distances. Hence, the representations learned by CYCLIP can be better used interchangeably than CLIP.

Table 4: Consistency score (%) trend for CLIP and CYCLIP across standard benchmarks . Top-k consistency score implies the fraction of times, the zero-shot predicted label in the text space is identical to the k-Nearest Neighbor predicted label in the image space (using the training dataset).

| | CIFAR-10 | | | | CIFAR-100 | | | | ImageNet1K | | | |
|---|---|---|---|---|---|---|---|---|---|---|---|---|
| | Top1 | Top3 | Top5 | Top10 | Top1 | Top3 | Top5 | Top10 | Top1 | Top3 | Top5 | Top10 |
| CLIP | 44.60 | 46.04 | 47.06 | 48.45 | 16.21 | 17.28 | 18.42 | 19.36 | 16.34 | 17.42 | 18.58 | 19.78 |
| CYCLIP | **48.81** | **50.89** | **52.30** | **53.71** | **20.43** | **21.96** | **23.18** | **24.31** | **19.20** | **20.31** | **21.95** | **23.94** |
| %GAIN | +8.6 | +9.5 | +10.0 | +9.8 | +20.7 | +21.3 | +20.5 | +20.4 | +14.9 | +14.2 | +15.4 | +17.4 |

## 4.2 Fine-grained and Coarse-grained Performance

In §3.1, we observed that CYCLIP outperforms CLIP on zero-shot transfer across various datasets. We perform an error analysis investigating both models' coarse and fine-grained classification performance to understand the transfer phenomena better. Given a hierarchical class structure dataset, coarse-grained classification differentiates between high-level (parent) classes, i.e., zero-shot classification into aquatic mammals and fish. The fine-grained classification task focuses on differentiating low-level (child) classes, i.e., zero-shot classification into a dolphin, otter, and seal (subclasses of aquatic mammals). We perform this analysis on the CIFAR-100, ImageNet1K, ImageNetV2, ImageNetSketch, ImageNet-A, and ImageNet-R datasets.

Formally, we consider a test set of $N$ image-subclass-superclass triplets, $\{I_j, C_j, P_j\}_{j=1}^N$, where $I_j$, $C_j$, $P_j$ represent the image, the subclass (child) and superclass (parent) respectively. The image embedding $I_j^e \in \mathbb{R}^d$ is obtained as described in §2, and the subclass embedding $C_j^e \in \mathbb{R}^d$ and superclass embedding $P_j^e \in \mathbb{R}^d$ are obtained as described in §3.1. Let the total number of superclasses and subclasses in the dataset be $n_\mathrm{p}$ and $n_\mathrm{c}$ , respectively. Further, let $F$ be a unique mapping from a subclass to the superclass, and $G$ denote the inverse mapping from a superclass to the set of subclasses i.e. $\forall P \in \{1, \ldots, n_\mathrm{p}\}$, $G(P) = \{C : F(C) = P \text{ and } C \in \{1, \ldots, n_\mathrm{c}\}\}$. Under this setup, the fine-grained and coarse-grained accuracies are defined as:

$$\text{Fine-grained Accuracy} = \frac{1}{N} \sum_{j=1}^N \mathbb{1}\left[\underset{C \in G(P_j)}{\mathrm{argmax}} \langle I_j^e, C\rangle = C_j\right] \qquad (6)$$

$$\text{Coarse-grained Accuracy} = \frac{1}{N} \sum_{j=1}^N \mathbb{1}\left[\underset{C \in \{1,\ldots,n_\mathrm{c}\}}{\mathrm{argmax}} \langle I_j^e, C\rangle \in G(P_j)\right] \qquad (7)$$

In Figure 3 we visualize how CLIP and CYCLIP compare with each other on the above metrics. The difference between the zero-shot performance of CYCLIP and CLIP is much more significant for coarse-grained classification than fine-grained classification across all the datasets. This observation indicates that concept-level knowledge is better captured in CYCLIP compared to CLIP. The drastic difference in the coarse-grained performance of CYCLIP and CLIP may be attributed to the rigid separation that the default cross-entropy loss in CLIP enforces between the positive pairs and negative pairs, which might degrade performance when some pairs in the negative batch belong to a similar entity. However, CYCLIP does not suffer from this problem as much because it poses cycle constraints on the overall geometry of all the data pairs rather than forcing a rigid separation.

## 4.3 Alignment and Uniformity on the Unit Hypersphere

[58] argues that contrastive learning directly optimizes for (a) alignment (closeness) of the representations of the positive pairs and (b) uniformity (coverage) of the representation space on the unit hypersphere. We extend these properties for multimodal contrastive representation learning as:

$$\text{Alignment} = \frac{1}{N} \sum_{j=1}^N \langle I_j^e, T_j^e\rangle \qquad \text{Uniformity} = \log\left(\frac{1}{N(N-1)} \sum_{j=1}^N \sum_{k=1, j\neq k}^N e^{-\langle I_j^e, T_k^e\rangle}\right) \qquad (8)$$

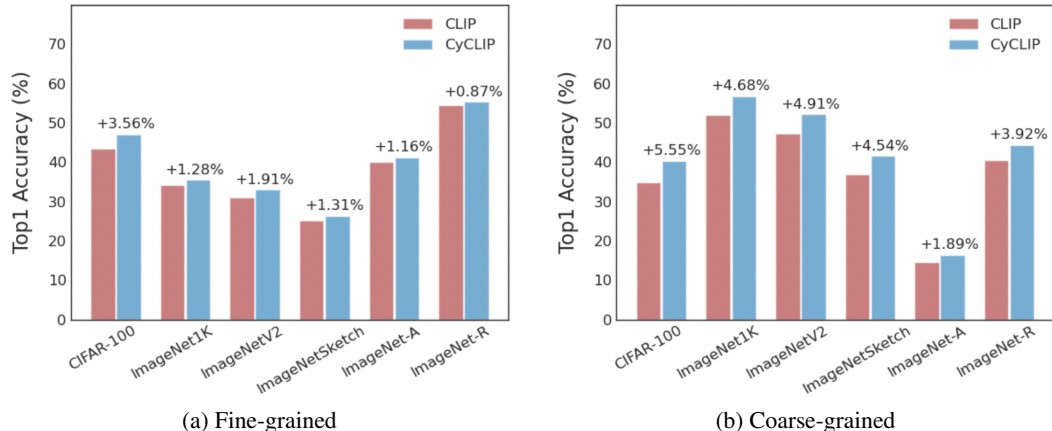

| (a) Fine-grained | (b) Coarse-grained |

Figure 3: The gap between the performances of CLIP and CYCLIP is much larger in coarse-grained scenario highlighting better entity-level knowledge representation in CYCLIP.

We desire our models to achieve high alignment and uniformity scores so that the image-text representations are close for the matched pairs and better spread over the unit hypersphere for different categories. We analyze the effect of cross-modal and in-modal consistency on the alignment and uniformity of the shared representations. For this, we train two ablated versions of CYCLIP, 1) C-CYCLIP with only cross-modal consistency component i.e. $\lambda_1 = 0, \lambda_2 = 0.5$, and 2) I-CYCLIP with only in-modal consistency component i.e. $\lambda_1 = 0.5, \lambda_2 = 0$ (in Eq. 5). We design proxy captions for classes as discussed in §3.1 to act as text embeddings. We present the results in Table 5.

Table 5: Alignment and Uniformity values for CLIP and Cyclic CLIP models. We abbreviate Alignment by A, Uniformity by U, and Zero-shot Top1 classification accuracy (%) by ZS-Top1.

| Model | CIFAR-10 | | | CIFAR-100 | | | ImageNet1K | | |
|---|---|---|---|---|---|---|---|---|---|
| | A | U | ZS-Top1 | A | U | ZS-Top1 | A | U | ZS-Top1 |
| CLIP | 0.36 | -0.27 | 46.54 | 0.36 | -0.25 | 18.69 | 0.39 | -0.18 | 20.03 |
| CYCLIP | 0.36 | -0.34 | 51.45 | 0.37 | -0.33 | 23.15 | 0.38 | -0.32 | **22.08** |
| I-CYCLIP | **0.60** | -0.57 | 50.97 | **0.60** | -0.57 | 22.35 | **0.61** | -0.55 | 21.21 |
| C-CYCLIP | 0.05 | **-0.02** | **55.52** | 0.06 | **-0.02** | **25.49** | 0.07 | **-0.02** | 21.73 |

We observe that I-CYCLIP learns representations that are better aligned in the representation space; however, they do not cover the hypersphere uniformly. The representations learned by C-CYCLIP are more uniformly spread but poorly aligned compared to I-CYCLIP. In this light, the components of CYCLIP can be seen to encourage a balance of good alignment and uniformity. Further, we find that CLIP is more uniform than CYCLIP in all datasets, but contrary to prior beliefs, this does not translate to improved downstream performance. C-CYCLIP has the best downstream zero-shot performance for CIFAR-10 and CIFAR-100 despite its poor alignment score. Further, all 3 variants of CYCLIP outperform CLIP on all 3 datasets, with CYCLIP performing the best on ImageNet1K.

## 4.4 Image-Text Retrieval

We evaluate the effectiveness of the proposed method on the cross-modal (image to text and text to image) retrieval downstream task in the zero-shot as well as fine-tuned settings. We consider the standard benchmark datasets: Flickr30K [42] and MSCOCO [8]. We assess our models on the test set of Flickr30K (1K) and MSCOCO (5K) obtained from the well-known Karpathy [30] split. Both the datasets contains 5 paired captions per image that makes text retrieval per image more easier than image retrieval per caption. We confirm the same in our results below. We perform fine-tuning on the Karpathy's training split with the batch size of 48. We fine-tune on Flick30K for 10 epochs and MSCOCO for 5 epochs. All the other hyperparameters are identical to that of pre-training.

Table 6: Zero-shot and fine-tuned cross-modal image-text retrieval (text-to-image and image-to-text) results of CLIP and CYCLIP on Flick30K and MSCOCO datasets.

| | | Flickr30K (1K) | | | | | | MSCOCO (5K) | | | | |
| | | Text Retrieval | | | Image Retrieval | | | Text Retrieval | | | Image Retrieval | | |
| | | R@1 | R@5 | R@10 | R@1 | R@5 | R@10 | R@1 | R@5 | R@10 | R@1 | R@5 | R@10 |
|---|---|---|---|---|---|---|---|---|---|---|---|---|---|
| Zero-shot | CLIP | 88.2 | 93.9 | 95.8 | 29.9 | 57.2 | 68.0 | 82.1 | 85.6 | 87.8 | 8.4 | 19.5 | 26.6 |
| | CYCLIP | 88.1 | 93.7 | 95.9 | **30.9** | 57.8 | **69.1** | 82.1 | 85.6 | 87.7 | 8.6 | 20.0 | 27.0 |
| Fine-tuned | CLIP | 91.9 | 97.0 | 98.0 | 46.3 | 74.7 | 83.6 | 83.2 | 87.6 | 90.0 | 10.6 | 23.9 | 31.3 |
| | CYCLIP | 92.3 | 97.0 | 98.4 | **47.3** | **76.6** | **85.4** | 83.2 | 87.8 | 90.3 | **11.4** | **25.8** | **33.4** |

Table 6 presents our cross-modal image-text retrieval results for CLIP and CYCLIP. In the zero-shot setting, we find that CYCLIP marginally outperforms CLIP on the image retrieval task on both the datasets. The relatively lower performance of both CLIP and CYCLIP in the zero-shot setting may be attributed to the more complicated nature of the two datasets where the models are expected to find similarities between the image and text at multiple resolutions as opposed to image classification where there is mostly single object to be matched with a simpler caption. It is not clear as to what distinctions in the raw input and text space are reflected in the embedding space too. Hence, we perform fine-tuning on both the datasets to better inform our models of the downstream datasets. In the fine-tuning setting, we find that the performance of both the models increases across both the datasets. However, we observe clear benefits of the soft consistent regularization on the image retrieval results for both the datasets.

## 4.5   CYCLIP preserves the Effective Robustness of CLIP

[36] shows that there is a strong correlation between the in-distribution and out-of-distribution generalization of the models trained on ImageNet1K, as illustrated by the linear fit (red) in Figure 4. Ideally, any model that does not undergo distribution shift would fall on the $y = x$ trendline (black). For other models, the deviations of the models from this ideal fit indicate their effective robustness. Previously, [45] showed that the zero-shot CLIP classifier trained on $400M$ image-text pairs improves effective robustness significantly compared to prior approaches to robustness. Subsequently, [28] demonstrated that CLIP models trained at small scales also exhibit high effective robustness that allows them to be used as a proxy to study the robustness properties of CLIP.

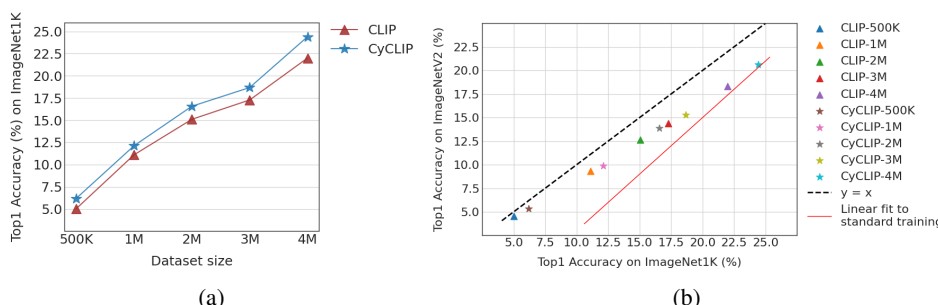

Figure 4: Effect of varying the training dataset size on (a) Classification accuracy on ImageNet1K and (b) Effective Robustness on ImageNetV2.

We evaluate the effect of cyclic consistency on effective robustness. We trained 4 CLIP and CYCLIP models, varying the training dataset sizes from $500K$ to $4M$ image-text pairs from the CC3M + CC12M datasets. In Figure 4, (a) we observe that for all training data sizes, CYCLIP shows a significant improvement over CLIP, showcasing its effectiveness in a diverse set of data regimes. Further, Figure 4 (b) shows that CYCLIP lies way above the baseline trend and preserves the effective robustness of CLIP.

# 5   Related Work

Our work fits into the broader theme of unsupervised pretraining with multiple modalities and has been successfully applied for learning representations of modalities such as images, text, and speech [2, 15, 1, 59, 43, 34]. Similar to the unimodal setting, two predominant approaches for multimodal pretraining are contrastive and generative, as described below.

**Contrastive Representation Learning:** Contrastive learning was originally proposed for self-supervised representation learning in the unimodal context where the embeddings of a sample are brought closer to an augmented version of the sample. In contrast, the embeddings are pushed away for other samples, and their augmentation [11, 51, 39, 55, 21, 7, 16, 40, 66, 23, 18]. [63] and [3] impose additional constraints to remove redundancies and prevent dimensional collapse in the visual representations. Recently, contrastive learning has also been used to learn robust representations of the multimodal data [62, 47]. Many works use additional losses to imbibe extra supervisory multimodal knowledge during the training process [54, 65, 64, 14, 35]. In this work, we focus on having cyclic consistency in addition to the contrastive loss to learn more robust image-text representations.

**Contrastive Language-Image Pretraining:** CLIP [44], ALIGN [29] and BASIC [41] have enjoyed great success in extending contrastive learning to paired image-text data, with impressive zero-shot classification and robustness performance. These works have been further extended recently to include visual self-supervision [37], additional nearest neighbor supervision [33], and utilization of unpaired data [56]. Our work complements much of this literature as it identifies consistency regularizers that can be augmented to the learning objective of the above works.

**Generative Representation Learning:** Generative models have been applied for learning representations of multimodal data [60, 53]. In particular, [67, 61, 10] proposed a notion of cyclic consistency for learning from unpaired multimodal data using GANs [17], which was extended later to normalizing flows [20, 19]. While these works focus on regularizing a generative mapping between modalities, our notion of cycle consistency applies to embeddings learned via a contrastive framework.

# 6   Conclusion

We presented CYCLIP, a framework for cycle consistent multimodal representation learning for image and text modality. The main benefits of CYCLIP stem from including cross-modal consistency and in-modal consistency regularizers to prevent inconsistent inference in the image and text spaces. Empirically, we show that CYCLIP performs much better than CLIP on zero-shot classification and is more robust on benchmarks for distributional robustness. We also showed that the representations learned by CYCLIP are more consistent than CLIP and better capture concept-level knowledge, as evidenced by our analysis of fine-grained and coarse-grained accuracies.

We believe this work can motivate further studies on understanding the geometry of the representation spaces learned via the contrastive objective applied to paired multimodal data and, in particular, identify conditions and regularization strategies under which the learned representations are synergistic across the various modalities for downstream applications.

One important future direction and a current limitation is scaling CYCLIP to larger datasets. While we do not possess the resources for this study, it is imperative to study the extent to which the benefits of cycle consistency remain at the scale on which the original CLIP was trained (400M image-text pairs). Finally, for real-world deployment of CLIP and their variants, such as CYCLIP, we need to be cautious about amplifying societal biases as these models are trained on large uncurated datasets scraped from the web [9]. Additionally, it is easy to add malicious data to the web, which poses a severe security threat [5]. Alleviating such harms is an important and active area of research.

## Acknowledgements

This research is supported by an Adobe Data Science Research Award for Aditya Grover. We would like to thank the IDRE's Research Technology group for the GPU computing resources on the UCLA Hoffman2 Cluster. We also want to thank Tung Duc Nguyen, Satvik Mashkaria, Siddarth Krishnamoorthy, Varuni Sarwal, and Ashima Suvarna for their helpful suggestions.

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
