# A   Additional Results

In addition to CYCLIP described in §2, we train two more instantiations of it by keeping either of the two consistency regularizers active in the loss objective (Eq. 5). The instantiation trained by setting $\lambda_1 = 0$ and $\lambda_2 = 0.5$ is termed as C-CYCLIP as only cross-modal consistency regularizer term is added to the loss objective. Similarly, we get I-CYCLIP where only in-modal consistency regularizer is added to the loss by setting $\lambda_1 = 0.5$ and $\lambda_2 = 0$. We evaluate C-CYCLIP and I-CYCLIP on most of the experiments discussed in the main text to understand their zero-shot transfer ability on standard datasets and robustness to natural distribution shifts.

## A.1   Zero-shot Transfer

Table 7 presents our results of the zero-shot transfer experiment described in §3.1. We find that CYCLIP outperforms its sub-variants and the CLIP model on the ImageNet1K dataset. Interestingly, we observe that the C-CYCLIP model performs the best amongst all models on CIFAR-10 and CIFAR-100. Further, we notice that all the versions of CYCLIP (the last three rows of Table 7) are better than CLIP across all the datasets. This indicates that jointly improving the geometry of the learned image and text representations using cyclic consistency regularizers is practical for improved zero-shot transfer performance on the image classification task.

Table 7: Zero-shot TopK classification accuracy (%) where K$\in \{1, 3, 5\}$

|  | CIFAR-10 | | | CIFAR-100 | | | ImageNet1K | | |
|---|---|---|---|---|---|---|---|---|---|
|  | Top1 | Top3 | Top5 | Top1 | Top3 | Top5 | Top1 | Top3 | Top5 |
| CLIP | 46.54 | 78.22 | 91.16 | 18.69 | 34.72 | 43.97 | 20.03 | 33.04 | 39.35 |
| CYCLIP | 51.45 | 79.57 | **91.80** | 23.15 | 41.46 | 50.66 | **22.08** | **35.98** | **42.30** |
| C-CYCLIP | **55.52** | **80.93** | 90.60 | **25.49** | **44.82** | **54.10** | 21.74 | 35.48 | 41.96 |
| I-CYCLIP | 50.97 | 79.51 | 90.75 | 22.35 | 39.25 | 48.45 | 21.22 | 34.72 | 41.05 |

## A.2   Natural Distribution Shifts

We evaluate the performance of the ablated CYCLIP models on natural distribution shift benchmarks to evaluate their distributional robustness and present our results in Table 8. We find that all the CYCLIP models (the last three rows) outperform CLIP by a large margin across all the datasets. Interestingly, we observe that C-CYCLIP performs the best on three of the four natural distribution shift datasets (last three columns). Among the CYCLIP models, I-CYCLIP performs the worst across all the datasets. This indicates that having just in-modal consistency is not enough to preserve the distributional robustness on the traditional datasets.

Table 8: Zero-shot classification on Natural Distribution Shifts (%)

|  | ImageNetV2 | | | ImageNetSketch | | | ImageNet-A | | | ImageNet-R | | |
|---|---|---|---|---|---|---|---|---|---|---|---|---|
|  | Top1 | Top3 | Top5 | Top1 | Top3 | Top5 | Top1 | Top3 | Top5 | Top1 | Top3 | Top5 |
| CLIP | 16.91 | 29.28 | 34.99 | 10.37 | 19.15 | 24.20 | 4.23 | 11.35 | 16.88 | 24.32 | 39.69 | 47.20 |
| CYCLIP | **19.22** | **32.29** | **38.41** | 12.26 | 22.56 | 28.17 | 5.35 | 13.53 | 19.51 | 26.79 | 42.31 | 50.03 |
| C-CYCLIP | 18.65 | 31.81 | 38.29 | **12.77** | **22.75** | **28.26** | **5.59** | **14.65** | **21.35** | **27.99** | **43.66** | **50.99** |
| I-CYCLIP | 18.40 | 30.63 | 36.82 | 10.87 | 20.17 | 25.49 | 4.75 | 11.84 | 16.92 | 24.55 | 38.71 | 45.68 |

## A.3   Linear Probe

We perform linear probing to assess the performance of CLIP and CYCLIP models in the presence of extra in-domain and extra in-modality supervision, i.e., with access to the training datasets for the visual classification task. We first get the image embeddings of the training data points from our trained visual encoder (ResNet-50), followed by training a learnable classifier (linear mapping).

Further, we train our models from scratch for 32 epochs on a single RTX2080Ti GPU with a batch size of 16 and an initial learning rate of 0.005 with cosine scheduling. We use a weight decay of 0.01 for the non-bias parameters of the linear layer.

We experiment with 14 visual classification datasets, details of which are present in Table 9. We present our linear probing results in Table 10. In particular, we observe that the CYCLIP models marginally outperform the CLIP model on all the datasets except Flowers102 and OxfordIIITPet.

Table 9: Training data size, Testing data size, and the number of classes for different datasets used for Linear Probe evaluation.

| Dataset | Classes | Train size | Test size |
|---|---|---|---|
| Caltech101 | 102 | 3060 | 6084 |
| CIFAR-10 | 10 | 50000 | 10000 |
| CIFAR-100 | 100 | 10000 | 10000 |
| DTD | 47 | 3760 | 1880 |
| FGVCAircraft | 100 | 6667 | 3333 |
| Flowers102 | 102 | 2040 | 6149 |
| Food101 | 101 | 75750 | 25250 |
| GTSRB | 43 | 26640 | 12630 |
| ImageNet1K | 1000 | 50000 | 50000 |
| OxfordIIITPet | 37 | 3680 | 3669 |
| RenderedSST2 | 2 | 6920 | 1821 |
| StanfordCars | 196 | 8144 | 8041 |
| STL10 | 10 | 5000 | 8000 |
| SVHN | 10 | 73257 | 26032 |

Table 10: Transfer CLIP and CYCLIP to 14 downstream visual datasets using linear probing. For training ImageNet1K, we use a random subset of 50K images from its original training dataset. Results have been averaged over 5 different seeds used for training the linear classifier.

| | Caltech101 | CIFAR-10 | CIFAR-100 | DTD | FGVCAircraft | Flowers102 | Food101 | GTSRB | ImageNet1K | OxfordIIITPet | RenderedSST2 | StanfordCars | STL10 | SVHN | Average |
|---|---|---|---|---|---|---|---|---|---|---|---|---|---|---|---|
| CLIP | 79.80 | 78.26 | 54.85 | 59.02 | 28.00 | **83.50** | 54.44 | 69.72 | 35.93 | **57.66** | 53.82 | 20.00 | 89.23 | 47.28 | 57.96 |
| CYCLIP | 80.33 | 76.98 | 55.74 | **63.44** | 27.86 | 82.96 | 54.96 | **71.70** | 37.12 | 56.82 | 53.74 | **22.14** | **90.10** | **48.01** | 58.71 |
| C-CYCLIP | **80.83** | **79.15** | **56.15** | 61.13 | **28.25** | 82.14 | **56.68** | 70.33 | **37.81** | 57.31 | **56.33** | 22.05 | 90.05 | 46.81 | **58.93** |
| I-CYCLIP | 80.73 | 77.77 | 55.48 | 61.87 | 27.46 | 81.85 | 54.00 | 68.26 | 36.86 | 57.44 | 53.60 | 20.83 | 89.95 | 46.89 | 58.07 |

## A.4  Consistency in Image and Text Spaces

Table 11 presents the results for the consistency score defined in Equation 2. We find that C-CYCLIP is the most consistent model despite missing the in-modal consistency regularizer term. Additionally, we observe that all the CYCLIP models (last three rows) are more consistent than the CLIP model across all the datasets. Hence, the explicit geometrization of the representations helps in improving the consistency of the image and text spaces.

## A.5  Fine-grained and Coarse-grained Performance

In §4.2, we described a novel fine-grained and coarse-grained analysis to investigate the zero-shot transfer phenomena better. We have 1000, 200, and 200 subclasses distributed across 67, 59, and 49 superclasses for the ImageNet1K/V2/Sketch, ImageNet-A, and ImageNet-R datasets, respectively.

Table 11: Consistency score (%) trend for CLIP and CYCLIP across standard benchmarks . Top-k consistency score implies the fraction of times, the zero-shot predicted label in the text space is identical to the k-Nearest Neighbor predicted label in the image space (using the training dataset).

| | CIFAR-10 | | | | CIFAR-100 | | | | ImageNet1K | | | |
|---|---|---|---|---|---|---|---|---|---|---|---|---|
| | Top1 | Top3 | Top5 | Top10 | Top1 | Top3 | Top5 | Top10 | Top1 | Top3 | Top5 | Top10 |
| CLIP | 44.60 | 46.04 | 47.06 | 48.45 | 16.21 | 17.28 | 18.42 | 19.36 | 16.34 | 17.42 | 18.58 | 19.78 |
| CYCLIP | 48.81 | 50.89 | 52.30 | 53.71 | 20.43 | 21.96 | 23.18 | 24.31 | 19.20 | 20.31 | 21.95 | 23.94 |
| C-CYCLIP | **53.47** | **56.56** | **57.84** | **59.27** | **22.72** | **24.03** | **25.85** | **26.95** | **20.02** | **21.30** | **23.06** | **24.75** |
| I-CYCLIP | 48.34 | 49.98 | 51.27 | 52.55 | 20.32 | 21.32 | 22.56 | 23.65 | 18.82 | 19.81 | 21.33 | 22.92 |

Figure 5 illustrates the distribution of the subclasses across the superclasses for these ImageNet benchmarks. For the CIFAR-100 dataset, we have 100 subclasses uniformly distributed across 20 superclasses, i.e., 5 subclasses per superclass[2].

Figure 6 illustrates the performance of CLIP and CYCLIP models across the traditional datasets and their natural distribution shifted variants. In tandem with our findings in §4.2, we observe that even ablated CYCLIP models, I-CYCLIP and C-CYCLIP outperform the CLIP model on the coarse-grained and fine-grained classification metric. Additionally, the margin of improvement is larger in the case of coarse-grained analysis than the fine-grained one. This highlights that all the CYCLIP variants are better at capturing entity-level knowledge than CLIP in their joint representations. We also find that CYCLIP captures more entity-level knowledge than C-CYCLIP in ImageNet1K and ImageNetV2, datasets containing natural images belonging to large number of categories.

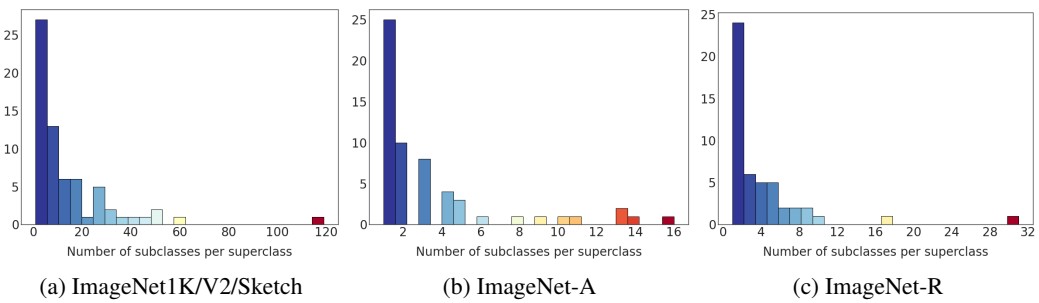

(a) ImageNet1K/V2/Sketch    (b) ImageNet-A    (c) ImageNet-R

Figure 5: Distribution of superclasses across different number of subclasses. ImageNet1K/V2/Sketch and ImageNet-A/R have 1000 and 200 subclasses respectively.

## A.6 Zero-shot Transfer: Additional Datasets

We present the results for zero-shot transfer downstream task, described in §3.1 in Table 12 across additional image classification datasets. The list of additional datasets is almost identical to the one used for linear probing §10. We include an additional dataset, *SUN397*, with 108754 images split across 397 classes. Further, we do not include some visual datasets as their labels were not amenable to constructing a simple proxy caption. For instance, *GTSRB* is a visual dataset with German traffic signs where a label is decided by the associated color, shape, and sign ID. Likewise, the images in the FGVCAircraft dataset are described by multiple attributes, and those in the RenderedSST2 dataset are described by positive/negative sentiments, hence not relevant in this setting. Furthermore, we present our results on a cleaner test set of CIFAR-10, CIFAR100, and ImageNet1K [**?** ]. We find that CYCLIP outperforms CLIP on the Top-1 classification accuracy with an average gain of 10.6% across all the datasets.

---

[2]https://www.cs.toronto.edu/ kriz/cifar.html

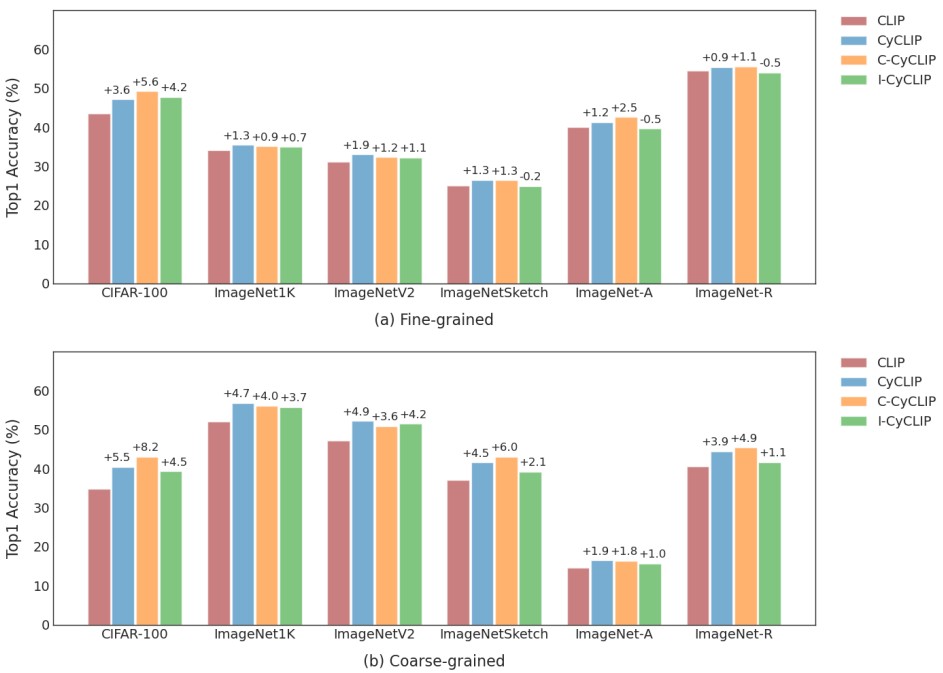

Figure 6: The gap between the performances of CLIP and our CyCLIP variants is much larger in coarse-grained scenario highlighting better entity-level knowledge representation in CyCLIPs.

Table 12: Zero-shot Top1 classification accuracy (%) across a battery of visual datasets. For the datasets marked with (*) we use their error-free labels instead of the original labels in their test sets.

| | Caltech101 | CIFAR-10* | CIFAR-100* | DTD | Flowers102 | Food101 | ImageNet1K* | OxfordIIITPets | StanfordCars | STL10 | SVHN | SUN397 | Average |
|---|---|---|---|---|---|---|---|---|---|---|---|---|---|
| CLIP | 52.6 | 46.3 | 18.4 | 14.7 | **13.7** | 13.3 | 20.2 | 2.2 | 1.2 | 83.5 | 7.1 | 39.6 | 26.1 |
| CYCLIP | **60.7** | **50.8** | **22.0** | **19.2** | 10.9 | **14.0** | **22.5** | **2.3** | **1.6** | **86.1** | **13.3** | **42.7** | **28.8** |
| Gain (%) | 15.4 | 9.8 | 19.1 | 30.0 | -20.4 | 5.0 | 11.8 | 7.3 | 34.4 | 3.1 | 86.3 | 7.7 | 10.6 |

# B   Additional Discussion

For the models pre-trained on the web-crawled data, one might argue against using the in-modal and cross-modal regularization as the image-text paired data is noisy where the caption may not fully describe the image at all resolutions. However, in an unsupervised setting, it is not pronounced as to what extent these distinctions should be reflected in the embedded representations.

As an example, we might think of the potential harm in bringing the embeddings of the two seemingly different captions ('A person playing with a dog on a beach', 'A person playing with a cat in a room') describing similar-looking images (with 'dog, person, beach, ball, sky', with 'cat, person, room') close using the in-modal consistency constraint. However, if the downstream task is to classify cats vs dogs, then the presence of any other objects in the images is a spurious correlation to be ignored. In such cases, we would desire the in-modal distance between text and image embeddings to be similar for consistent predictions in both modalities. Hence, the effectiveness of any information captured in a modality depends on the downstream task.

Moreover, one could make a similar counter-argument about the text and image modalities being inherently different against CLIP's contrastive loss. It weighs each image and text pair equally, even

though some text captions might be more descriptive about the images than others and therefore should be assigned a higher weight. From a practical standpoint, our evidence suggests that soft consistency regularization in the form of additional loss terms, as in Equation 5, can be generally helpful for the downstream tasks and domain settings of interest. The relative gains can indeed be different across tasks and domains. For example, in Figure 3, while our regularizers lead to gains in fine and coarse-grained classification, we noted that the relative gains are much higher for coarse-grained settings due to the improved consistency.

## C  Pretraining and Implementation details

### C.1  Dataset

Conceptual Captions 3M (CC3M) [52] is an open-source dataset consisting of approximately 3.3M image-caption pairs scarped from the web. The dataset (including train/validation split) is made available by Google [3]. Due to the broken image URLs, we use a subset of 2,631,703 image-caption pairs for pre-training. Further, Conceptual 12M (CC12M) [6] is a noisy extension to the CC3M dataset and contains approximately 12M image-caption pairs, covering a more diverse set of visual concepts. For the dataset ablation experiments in Figure 4 (a), with a data size of 3M pairs, we extend our CC3M dataset using a random subset of image-caption pairs (368,297 pairs) from CC12M [4].

### C.2  Model architecture

Our models use the same architecture as the original CLIP model presented in [44] with a ResNet-50 image encoder (38,316,896 parameters) and a transformer-based text encoder with a projection layer (63,690,240 parameters) to match the image embedding dimension of 1024. We use a weight decay for all the parameters during training, except for batch/layer norm, bias, and logit scale parameters.

### C.3  Hyperparameter settings

The hyperparameters for the best-performing configuration were inspired from the original paper on CLIP [44] and mainly taken from mlfoundations[5] repository. However, in our experiments, we use a batch size of 128 due to limited computation resources. To avoid hurting the performance of our models, we train our models for 64 epochs. For the cyclic consistency hyperparameters $(\lambda_1, \lambda_2)$, we used a combination of zoom grid search and manual tuning on a small subset of 480K image-text pairs with training on 16 epochs and optimizing the contrastive loss on the Conceptual Captions validation set of 13,156 examples. We found that CyCLIP loss is not very sensitive in the non-zero parameter space between 0 and 1. Therefore, we choose a more simplistic setup with both the values as 0.25. We trained the CLIP and CyCLIP models on Azure with 4x Tesla-V100-SXM2-16GB GPUs and 24x CPUs for approximately 84 hours each. The hyperparameter settings are shown in table 13.

Table 13: Hyperparameters used for training the CLIP models

| Hyperparameter | Value |
| --- | --- |
| Embedding dimension | 1024 |
| Logit scale range | 0 to 4.6052 |
| Epochs | 64 |
| Batch size | 128 |
| Learning rate | 0.0005 |
| Adam beta1 | 0.9 |
| Adam beta2 | 0.99 |
| Adam weight decay | 0.1 |
| Scheduler | Cosine |
| Learning rate warmup steps | 10000 |

---

[3]https://ai.google.com/research/ConceptualCaptions/download/
[4]https://github.com/google-research-datasets/conceptual-12m
[5]https://github.com/mlfoundations/open_clip