# OpenReview forum: "CyCLIP: Cyclic Contrastive Language-Image Pretraining"
_NeurIPS.cc/2022/Conference — NeurIPS 2022 Accept_

### Official Review · Reviewer_ra5C · 2022-07-10

**Rating:** 7
**Confidence:** 4
**Soundness:** 3 good
**Presentation:** 4 excellent
**Contribution:** 4 excellent

**Summary:**

The paper analyzes the representation space of CLIP and identifies that the image and text distances are inconsistent — the concepts closest to the image in an image-text pair can be very different from the concepts closest to the text. They then propose losses to fix this distortion, and outperform CLIP trained on a similar amount of data.



**Questions:**

Please see the weaknesses / strengths.

1) Can this method be easily combined with other xyzCLIP methods, for example, DeCLIP?

**Limitations:**

The authors have adequately addressed the limitations of their work.

**Strengths And Weaknesses:**

The paper points out a very interesting phenomenon in the representation space of CLIP, and proposes a simple method to fix the representation space. The method works quite well empirically. I find this particularly useful because it is orthogonal to many other recent approaches that build on the original CLIP formulation. It should be possible to combine this with any image-text contrastive learner. Overall, the method is presented clearly and convincingly.

There are two main weaknesses to the paper. The first is the size of the dataset. Would the advantages of this method disappear as the amount of data available increases? At 4M samples, CyCLIP is $2$% better than CLIP-4m on ImageNet1k. How would this change as the amount of data increases, let's say to 15M? Reproducing the full CLIP-400m results is obviously infeasible, and even a 15M dataset is nontrivial, so I think the scale of the evaluation in the paper is _probably_ okay. The paper makes an attempt to address this (Fig. 4).

The more serious weakness to me is the sparsity of the experimental evaluation of zero-shot classification. Zero-shot evaluation is computationally _extremely_ cheap. The zero-shot results are only included for 3 datasets, of which 2 are CIFAR-10/100. CIFAR-10/100 are very old at this point, and are known to contain many mislabeled samples in the test set [1]. I would like to see an evaluation for some other datasets which are more challenging and cleaner. I leave the choice of dataset up to the authors — the original CLIP paper contains many other datasets that can be used for purposes of comparison. A simple choice would be image-text retrieval, for example on the Flickr30K dataset.


[1] https://arxiv.org/abs/2103.14749

---

> ### Author Response · Authors · 2022-08-02
> **Author Response [1/N]**
>
> We would like to sincerely thank the reviewer for their diligent feedback. We are encouraged to find that the reviewer found our work (a) engaging in the space of representation learning, (b) a simple solution to a fundamental inconsistency problem, and (c) convincing. We are grateful to the reviewer for noticing that our method is particularly modular that can be combined with any image-text contrastive learner.
>
> > **The first is the size of the dataset. Would the advantages of this method disappear as the amount of data available increases?**
>
> We thank the reviewer for asking a very pertinent question. We were aware of this limitation in the original draft and mentioned it in our Conclusion section. We have very scarce resources for training models in the short rebuttal period, but we are trying our best to get larger scale results in the remainder of the rebuttal period. We'll update if and when we get those results.
>
> > **Can this method be easily combined with other xyzCLIP methods, for example, DeCLIP?**
>
> Yes, this regularization method can easily be combined with other CLIP variants like DeCLIP, SLIP, FILIP, ALBEF, FLAVA, etc., and other vision-language pretraining models. The in-modal and cross-modal cyclic consistency terms are reasonably generic and open interesting directions for testing this framework in different settings.

---

> > ### Author Response · Authors · 2022-08-02
> > **Author Response: Zero-shot (Additional Datasets) [2/N]**
> >
> > > **The more serious weakness to me is the sparsity of the experimental evaluation of zero-shot classification. Zero-shot evaluation is computationally extremely cheap. I would like to see an evaluation for some other datasets which are more challenging and cleaner.**
> >
> > Most of the related works referenced in our paper, such as DeCLIP and SLIP, primarily focus on evaluating their models on the Imagenet 1K benchmark. However, we include CIFAR-10 and CIFAR-100 as well to evaluate our models on benchmarks with fewer classes. Based on the reviewer’s suggestion, we compare the performance of CLIP and CyCLIP on more datasets as presented below. Additionally, we thank the reviewer for suggesting a good resource that highlights the issue of labeling errors in these benchmarks. Based on that, we have evaluated our models on the **corrected (cleaner)** versions of CIFAR10, CIFAR100, and ImageNet1K, also shown below. As expected, CyCLIP outperforms CLIP on most of the zero-shot benchmarks.
> >
> > |Zero-shot results |       |      |       |||||||
> > |------|----------------|--------------|---------------|------|------|------|------|------|-------|
> > |      |                |              |               |      |      |      |      |      |       |
> > |      |**Cal101**      | ||   **C10**   |      |      |      **C100** |      |       |
> > |      |**Top1**            |**Top3**          |**Top5**|**Top1**            |**Top3**          |**Top5**|**Top1**            |**Top3**          |**Top5**|
> > |CLIP  |52.58           |72.99         |77.81          |46.26 |78.08 |90.95 |18.43 |33.84 |42.73  |
> > |CyCLIP|60.68           |77.61         |80.79          |50.81 |79.15 |91.51 |21.95 |39.92 |48.95  |
> > |Gain %|**15.41**           |**6.33**          |**3.83**           |**9.84**  |**1.37**  |**0.62**  |**19.1**  |**17.97** |**14.56**  |
> > |      |                |              |               |      |      |      |      |      |       |
> > |      |**DTD**             |   |        |   **F102**    |      |      |   **F101**   |      |       |
> > |      |**Top1**            |**Top3**          |**Top5**|**Top1**            |**Top3**          |**Top5**|**Top1**            |**Top3**          |**Top5**|
> > |CLIP  |14.73           |26.48         |33.56          |13.71 |24.93 |29.69 |13.32 |25.65 |32.46  |
> > |CyCLIP|19.15           |32.08         |39.31          |10.91 |19.94 |24.21 |13.99 |25.28 |32.16  |
> > |Gain %|**30.01**           |**21.15**         |**17.13**          |-20.42|-20.02|-18.46|**5.03**  |-1.44 |-0.92  |
> > |      |                |              |               |      |      |      |      |      |       |
> > |      |**IN1K**| |   |    **OPet**  |      |      |    **SCars**  |      |       |
> > |      |**Top1**            |**Top3**          |**Top5**|**Top1**            |**Top3**          |**Top5**|**Top1**            |**Top3**          |**Top5**|
> > |CLIP  |20.16           |33.08         |39.25          |2.18  |9.18  |15.21 |1.22  |3.28  |5.45   |
> > |CyCLIP|22.54           |36.27         |42.44          |2.34  |8.39  |14.61 |1.64  |4.13  |6.34   |
> > |Gain %|**11.82**           |    **9.66**      |**8.12**           |**7.34**  |-8.61 |-3.94 |**34.43** |**25.91** |**16.33**  |
> > |      |                |              |               |      |      |      |      |      |       |
> > |      |**STL10**           |          |         | **SVHN**     |      |      |   **SUN397**   |      |       |
> > |      |**Top1**            |**Top3**          |**Top5**|**Top1**            |**Top3**          |**Top5**|**Top1**            |**Top3**          |**Top5**|
> > |CLIP  |83.49           |95.18         |98.33          |7.13  |26.99 |56.81 |39.63 |61.22 |69.33  |
> > |CyCLIP|86.11           |97.41         |99.51          |13.28 |40.98 |59.88 |42.69 |63.58 |71.68  |
> > |Gain %|**3.14**            |**2.34**          |**1.2**            |**86.26** |**51.83** |**5.4**   |**7.72**  |**3.85**  |**3.39** |
> >
> > *Abbreviations: Cal101-Caltech101, C10-CIFAR10, C100 - CIFAR100, F102 - Flowers102, F101 - Food101, IN1K - ImageNet1K, OPet-OxfordIIITPet, SCars - StanfordCars.*
> >
> > Further, we would like to bring attention to the experiments in Sections 3.2 and 4.2, where we compare CLIP and CyCLIP on ImageNet-A/R/V2/Sketch datasets that have shown to be very challenging for the ImageNet models. We showcase that CyCLIP outperforms CLIP on all these datasets on zero-shot evaluation, thus highlighting its robustness to natural distribution shifts compared to the CLIP model.

---

> > > ### Author Response · Authors · 2022-08-02
> > > **Author Response: Image-Text Retrieval [3/N]**
> > >
> > > > **A simple choice would be image-text retrieval, for example on the Flickr30K dataset.**
> > >
> > > As requested by the reviewer, we conducted zero-shot and fine-tuned cross-modal retrieval experiments on the standard Flickr30K 1K and MSCOCO 5K test set. For both the datasets, image-to-text retrieval requires each image to retrieve one of the five relevant captions in its top K closest predictions. In contrast, the text-to-image retrieval requires each caption to retrieve the correct image (only one possible) in its top K closest predictions. This explains why we observe lower image retrieval scores over text retrieval scores. While CLIP and CyCLIP are comparable on the easier text retrieval tasks, we observe that CyCLIP outperforms CLIP across both the datasets on the Image retrieval task in both the zero-shot and fine-tune cases.
> > >
> > > *Abbreviation: TR - Text Retrieval, IR - Image Retrieval*
> > >
> > >
> > > | Flickr30K (1K) | R@1 (TR) | R@5 (TR) | R@10 (TR) | R@1 (IR) | R@5 (IR) | R@10 (IR) |   |   |   |   |
> > > |-------------|----------|----------|-----------|----------|----------|-----------|---|---|---|---|
> > > | **Zero-shot**      |          |          |           |          |          |           |   |   |   |   |
> > > | CLIP           | 88.2     | 93.9     | 95.8      | 29.9     | 57.2     | 68        |   |   |   |   |
> > > | CyCLIP         | 88.1     | 93.7     | 95.9      | **30.9**     | 57.8     | **69.1**      |   |   |   |   |
> > > | **Fine-tuned**      |          |          |           |          |          |           |   |   |   |   |
> > > | CLIP           | 91.9     | 97       | 98        | 46.3     | 74.7     | 83.6      |   |   |   |   |
> > > | CyCLIP         | 92.3     | 97       | 98.4      | **47.3**     | **76.6**    | **85.4**      |   |   |   |   |
> > > |                |          |          |           |          |          |           |   |   |   |   |
> > >
> > > | MSCOCO (5K) | R@1 (TR) | R@5 (TR) | R@10 (TR) | R@1 (IR) | R@5 (IR) | R@10 (IR) |   |   |   |   |
> > > |-------------|----------|----------|-----------|----------|----------|-----------|---|---|---|---|
> > > |**Zero-shot**  |          |          |           |          |          |           |   |   |   |   |
> > > | CLIP        | 82.1     | 85.6     | 87.8      | 8.4      | 19.5     | 26.6      |   |   |   |   |
> > > | CyCLIP      | 82.1     | 85.6     | 87.7      | 8.6      | 20       | 27        |   |   |   |   |
> > > | **Fine-tuned**   |          |          |           |          |          |           |   |   |   |   |
> > > | CLIP        | 83.2     | 87.6     | 90        | 10.6     | 23.9     | 31.3      |   |   |   |   |
> > > | CyCLIP      | 83.2     | 87.8     | 90.3      | **11.4**     | **25.8**     | **33.4**      |   |   |   |   |
> > > |             |          |          |           |          |          |           |   |   |   |   |
> > >
> > > The relatively lower performance of both CLIP and CyCLIP in the zero-shot setting may be attributed to the more complicated nature of the two datasets where the models are expected to find similarities between the image and text at multiple resolutions as opposed to image classification where there is mostly single object to be matched with a simpler caption. It is not clear as to what distinctions in the raw input and text space are reflected in the embedding space too. Hence, we perform fine-tuning on both the datasets to better inform our models of the downstream datasets.
> > >
> > > We further create a text consistency metric that measures the proportion of the captions for which we retrieve the correct image and one of the four similar captions simultaneously over the whole dataset. This is similar to Equation 2 in our paper. We find that CyCLIP outperforms CLIP on fine tuning for both the datasets. The following table summarizes are results.
> > >
> > > |           | Flickr30K 1K | MSCOCO 5K |
> > > |-----------|--------------|-----------|
> > > | **Zero-shot** |              |           |
> > > | CLIP      | 20.9         | 2.8       |
> > > | CyCLIP    | **21.3**         | 2.8       |
> > > | **Fine-tuned** |              |           |
> > > | CLIP      | 35.4         | 3.8       |
> > > | CyCLIP    | **37.7**         | **4.1**       |

---

> > > > ### Comment · Reviewer_ra5C · 2022-08-05
> > > > **Response**
> > > >
> > > > The new experimental results look good, and I am raising my score. I encourage the authors to include these results in the camera ready.

---

> > > > > ### Author Response · Authors · 2022-08-05
> > > > > **(Update) Author Response: Larger Dataset Size [4/N]**
> > > > >
> > > > > > **The first is the size of the dataset. Would the advantages of this method disappear as the amount of data available increases?**
> > > > >
> > > > > As mentioned in our rebuttal, with our limited resources, we were able to complete the training of the CLIP and CyCLIP models on a larger dataset size of 4Million (CC3M + 1M from CC12M). The training hyperparameters are identical to the 3M runs.
> > > > >
> > > > > We present the zero-shot Top-1 classification accuracy results on the series of challenging datasets. We observe that CyCLIP outperforms CLIP across all the datasets showcasing better zero-shot ability as well as the robustness to natural distribution shift at a larger scale.
> > > > >
> > > > > | Top-1       | IN-1K | IN-V2 | IN-Sk | IN-A | IN-R |
> > > > > |-------------|-------|-------|-------|------|------|
> > > > > | CLIP (4M)   | 22.0  | 18.3  | 13.0  | 4.8  | 27.4 |
> > > > > | CyCLIP (4M) | 24.4  | 20.6  | 14.8  | 5.4  | 30.4 |
> > > > > | Gain (%)       | **11.1**  | **12.7**  | **13.6**  | **10.8** | **11.0** |
> > > > > |||||||||
> > > > >
> > > > > *Abbreviations: IN-1K - ImageNet-1K, IN-V2 - ImageNet V2, IN-Sk - ImageNet-Sketch, IN-A - ImageNet-A, IN-R - ImageNet-R*

---

### Official Review · Reviewer_KkEn · 2022-07-11

**Rating:** 7
**Confidence:** 5
**Soundness:** 4 excellent
**Presentation:** 4 excellent
**Contribution:** 4 excellent

**Summary:**

This paper provides the first analysis of inconsistency problem in contrastive learning of image-text models, and proposes new training objectives to mitigate the issue. Specifically, it demonstrates that within-modality similarities can be inconsistent with cross-modality ones, a potential area of improvement for existing models. Based on this observation, the author propose to add two regularization terms during training to alleviate the issue. Experimental results clearly show the performance gain of the method.

**Questions:**

Some questions/suggestions:
1. In section 2.2, do you have some good intuition/understanding why the model can still obtain good performance using text classifications while the consistency is bad?
2. I wonder if it should be N^2 instead of N in the normalization terms in Eq (3) and (4)?
3. How do you select your hyper-parameters?
4. Have you considered tasks other than image classification? e.g. how does the model perform on cross-model retrieval?

**Limitations:**

This has been covered by the paper.

**Strengths And Weaknesses:**

Overall, I found the paper interesting to read. The observation made is inspiring and proposed method does appear to be effective. There are, however, some areas that can be improved for a stronger paper (see below).

Strengths:
1. The inconsistency issue has never been identified before, to the best of my knowledge. I think this is an interesting and important issue that needs to be addressed properly.
2. The proposed methods are intuitive and easy to use. Experimental results clearly show its effectiveness.
3. The paper is well written, with plenty of inspiring analysis.

Weakness:
1. My major concern is the experimental setup. The experiment is conducted on small-scale CLIP models with small batch sizes. As a result, the performance is far away from the real CLIP or CLIP-like methods. It therefore leaves room for doubts on how the analysis and results generalize to larger scales.
2. Another point is that while inconsistency is a concerning issue, it will be nice to show some actual cases where it can hurt model performance, even on a curated synthetic setting.

---

> ### Author Response · Authors · 2022-08-02
> **Author Response [1/ N]**
>
> We thank the reviewer for finding our work interesting, as well as inspiring. We are motivated to find that the reviewer found our work (a) novel in terms of the idea of consistency, (b) intuitive and modular, and (c) well-written with inspiring analysis. We address the concerns brought up by the reviewer one by one and incorporate their suggestions.
>
> > **My major concern is the experimental setup. The experiment is conducted on small-scale CLIP models with small batch sizes. As a result, the performance is far away from the real CLIP or CLIP-like methods. It therefore leaves room for doubts on how the analysis and results generalize to larger scales.**
>
> We thank the reviewer for asking a very pertinent question. We were aware of this limitation in the original draft and mentioned it in our Conclusion section.
>
> We would like to bring the attention of the reviewer to the Section 4.4 Figure 4(a) where we try to partially answer this question by training 4 CLIP and CyCLIP models at various pre-training dataset sizes. We find that the CyCLIP outperforms CLIP across the whole range of pre-training corpus.
>
> Additionally, we have very scarce resources for training models in the short rebuttal period, but we are trying our best to get larger scale results in the remainder of the rebuttal period. We'll update if and when we get those results.
>
> > **Another point is that while inconsistency is a concerning issue, it will be nice to show some actual cases where it can hurt model performance, even on a curated synthetic setting.**
>
> We thank the reviewer for believing that inconsistency is a concerning issue in the multimodal settings. As an example of a downstream task, recent work [1] uses the CLIP’s text and image embeddings interchangeably, image embeddings during training and text embeddings using inference, to transfer to a new domain: 3D shapes. For example, the reviewer may think of the same setting as illustrated in Figure 1 of our paper with a simple change where we get a third modality (3D shape of a cat) instead of the test image of a cat. In this case, a framework like CLIP-forge would bring the representation of the 3-D shape of the cat close to the pre-trained image representation of the cat during the training; however, the pre-trained text representation of the cat may still be far from the representation of 3D rendition of a cat. This could lead to incorrect 3D generations during inference with a given text.
>
> Additionally, In section 4.1, we point out the consistency score differences between CLIP and CyCLIP.
>
> > **In section 2.2, do you have some good intuition/understanding why the model can still obtain good performance using text classifications while the consistency is bad?**
>
> The cross-modal contrastive loss in the original CLIP setting explicitly optimizes for increasing the similarity between paired text and image data. Hence, intuitively, it seems that the contrastive loss is well-aligned with how CLIP prompts are used for classification by comparing similarity scores, which explains why it gets good performance. However, no constraint forces the model to keep embeddings of similar images (or text) closer and distinct images (or text) further from each other. Consistency constraints further aid the model to generalize to unseen pairs in zero-shot settings by using mismatched and in-modal pairs to regularize the learned representation space.
>
> > **I wonder if it should be N^2 instead of N in the normalization terms in Eq (3) and (4)?**
>
> We thank the reviewer for taking a careful look at our equations. However, we use the normalization term as N only because there are N image and text pairs in a batch of size N.
>
> > **How do you select your hyper-parameters?**
>
> The hyperparameters for the CLIP model were mainly taken from https://github.com/mlfoundations/open_clip repository. However, in our experiments, we use a batch size of 128 due to limited computation resources. To avoid hurting the performance of our models, we train our models for 64 epochs. For the cyclic consistency hyperparameters (lambda1, lambda2), we used a combination of zoom grid search, and manual tuning on a small subset of 480K image-text pairs with training on 16 epochs and optimizing the contrastive loss on the Conceptual Captions validation set of 13156 examples. In general, we found that CyCLIP loss is not very sensitive in the non-zero parameter space between 0 and 1. Therefore, we choose a more simplistic setup with both the values as 0.25.
>
> **Reference:**
>
> [1] CLIP-Forge: Towards Zero-Shot Text-to-Shape Generation. https://openaccess.thecvf.com/content/CVPR2022/papers/Sanghi_CLIP-Forge_Towards_Zero-Shot_Text-To-Shape_Generation_CVPR_2022_paper.pdf

---

> > ### Author Response · Authors · 2022-08-02
> > **Author Response: Image-Text Retrieval [2/N]**
> >
> > > **Have you considered tasks other than image classification? e.g. how does the model perform on cross-model retrieval?**
> >
> > As requested by the reviewer, we conducted zero-shot and fine-tuned cross-modal retrieval experiments on the standard Flickr30K 1K and MSCOCO 5K test set. For both the datasets, image-to-text retrieval requires each image to retrieve one of the five relevant captions in its top K closest predictions. In contrast, the text-to-image retrieval requires each caption to retrieve the correct image (only one possible) in its top K closest predictions. This explains why we observe lower image retrieval scores over text retrieval scores. While CLIP and CyCLIP are comparable on the easier text retrieval tasks, we observe that CyCLIP outperforms CLIP across both the datasets on the Image retrieval task in both the zero-shot and fine-tune cases.
> >
> > *Abbreviation: TR - Text Retrieval, IR - Image Retrieval*
> >
> >
> > | Flickr30K (1K) | R@1 (TR) | R@5 (TR) | R@10 (TR) | R@1 (IR) | R@5 (IR) | R@10 (IR) |   |   |   |   |
> > |-------------|----------|----------|-----------|----------|----------|-----------|---|---|---|---|
> > | **Zero-shot**      |          |          |           |          |          |           |   |   |   |   |
> > | CLIP           | 88.2     | 93.9     | 95.8      | 29.9     | 57.2     | 68        |   |   |   |   |
> > | CyCLIP         | 88.1     | 93.7     | 95.9      | **30.9**     | 57.8     | **69.1**      |   |   |   |   |
> > | **Fine-tuned**      |          |          |           |          |          |           |   |   |   |   |
> > | CLIP           | 91.9     | 97       | 98        | 46.3     | 74.7     | 83.6      |   |   |   |   |
> > | CyCLIP         | 92.3     | 97       | 98.4      | **47.3**     | **76.6**    | **85.4**      |   |   |   |   |
> > |                |          |          |           |          |          |           |   |   |   |   |
> >
> > | MSCOCO (5K) | R@1 (TR) | R@5 (TR) | R@10 (TR) | R@1 (IR) | R@5 (IR) | R@10 (IR) |   |   |   |   |
> > |-------------|----------|----------|-----------|----------|----------|-----------|---|---|---|---|
> > |**Zero-shot**  |          |          |           |          |          |           |   |   |   |   |
> > | CLIP        | 82.1     | 85.6     | 87.8      | 8.4      | 19.5     | 26.6      |   |   |   |   |
> > | CyCLIP      | 82.1     | 85.6     | 87.7      | 8.6      | 20       | 27        |   |   |   |   |
> > | **Fine-tuned**   |          |          |           |          |          |           |   |   |   |   |
> > | CLIP        | 83.2     | 87.6     | 90        | 10.6     | 23.9     | 31.3      |   |   |   |   |
> > | CyCLIP      | 83.2     | 87.8     | 90.3      | **11.4**     | **25.8**     | **33.4**      |   |   |   |   |
> > |             |          |          |           |          |          |           |   |   |   |   |
> >
> > The relatively lower performance of both CLIP and CyCLIP in the zero-shot setting may be attributed to the more complicated nature of the two datasets where the models are expected to find similarities between the image and text at multiple resolutions as opposed to image classification where there is mostly single object to be matched with a simpler caption. It is not clear as to what distinctions in the raw input and text space are reflected in the embedding space too. Hence, we perform fine-tuning on both the datasets to better inform our models of the downstream datasets.
> >
> > We further create a text consistency metric that measures the proportion of the captions for which we retrieve the correct image and one of the four similar captions simultaneously over the whole dataset. This is similar to Equation 2 in our paper. We find that CyCLIP outperforms CLIP on fine tuning for both the datasets. The following table summarizes are results.
> >
> > |           | Flickr30K 1K | MSCOCO 5K |
> > |-----------|--------------|-----------|
> > | **Zero-shot** |              |           |
> > | CLIP      | 20.9         | 2.8       |
> > | CyCLIP    | **21.3**         | 2.8       |
> > | **Fine-tuned** |              |           |
> > | CLIP      | 35.4         | 3.8       |
> > | CyCLIP    | **37.7**         | **4.1**       |

---

> > > ### Author Response · Authors · 2022-08-05
> > > **(Update) Author Response: Larger Dataset Size [3/N]**
> > >
> > > > **My major concern is the experimental setup. The experiment is conducted on small-scale CLIP models with small batch sizes. As a result, the performance is far away from the real CLIP or CLIP-like methods. It therefore leaves room for doubts on how the analysis and results generalize to larger scales.**
> > >
> > > As mentioned in our rebuttal, with our limited resources, we were able to complete the training of the CLIP and CyCLIP models on a larger dataset size of 4Million (CC3M + 1M from CC12M). The training hyperparameters are identical to the 3M runs.
> > >
> > > We present the zero-shot Top-1 classification accuracy results on the series of challenging datasets. We observe that CyCLIP outperforms CLIP across all the datasets showcasing better zero-shot ability as well as the robustness to natural distribution shift at a larger scale.
> > >
> > > | Top-1       | IN-1K | IN-V2 | IN-Sk | IN-A | IN-R |
> > > |-------------|-------|-------|-------|------|------|
> > > | CLIP (4M)   | 22.0  | 18.3  | 13.0  | 4.8  | 27.4 |
> > > | CyCLIP (4M) | 24.4  | 20.6  | 14.8  | 5.4  | 30.4 |
> > > | Gain (%)       | **11.1**  | **12.7**  | **13.6**  | **10.8** | **11.0** |
> > > |||||||||
> > >
> > > *Abbreviations: IN-1K - ImageNet-1K, IN-V2 - ImageNet V2, IN-Sk - ImageNet-Sketch, IN-A - ImageNet-A, IN-R - ImageNet-R*

---

### Official Review · Reviewer_tnET · 2022-07-12

**Rating:** 7
**Confidence:** 2
**Soundness:** 3 good
**Presentation:** 3 good
**Contribution:** 2 fair

**Summary:**

This paper argues that the standard contrastive objective adopted in large-scale image-text pre-training (e.g., CLIP) may lead to inconsistency in learned representations for zero-shot image classification application. To improve the inconsistency, the authors propose to add two regularizers to standard contrastive objective to enforce both cross-modal consistency and in-modal consistency. Pre-training on CC3M, the proposed CyCLIP achieves better ZS performance, consistency scores and linear probing results on image classification than CLIP.

**Questions:**

- Would the added regularizers affect the performance on some applications other than image classification, for example image-to-text retrieval?
- It is also not very clear whether the current advantage of CyCLIP would hold when scaling up to larger pretraining corpus. To perform a fast validation, maybe the authors can add 1M image-text pairs from SBU dataset or even 12M data from CC12M as additional pre-training data.

**Limitations:**

Discussed in conclusion section.

**Strengths And Weaknesses:**


In general, the paper is clearly written, easy to follow. The proposed CyCLIP is simple, with just two more added regularizer losses to the standard contrastive loss, yet has shown more effective than CLIP on image classification tasks over different evaluation settings.

My main concern is about the assumptions made by these regularizer terms, and would love to discuss with the authors. In-modal consistency seem to assume the distances measure in image representation space is at the same scale as that in the text representation space. Despite the input image and text space maybe inherently different, the assumption may not be true when the text does not fully describe the image (which is the majority case in web-crawled data)..

- image-1: dog, person, beach, ball, sky
- text-1: A person playing with a dog on a beach
- image-2: cat, person, room
- text-2: A person playing with a cat in a room

In my personal opinion, for the above examples, the distances between the texts are much closer than the distances between images.

The symmetry of distance measures for cross-modal inconsistency also may not be valid.

- image-1: dog, person, room, ball
- text-1: A person playing a ball with the dog in a room
- image-2: cat, person, room, ball
- text-2: A person playing with a cat

If both images are the person playing a ball with the [dog/cat] in a room, then I would think <Image-1, text-2> may be greater than <image-2, text-1>. As the edit steps on text-1 to make it fully describe image-2, is just to change the word dog to cat, while more edits are needed on text-2, to make it describe image-1.

---

> ### Author Response · Authors · 2022-08-02
> **Author Response [1/N]**
>
> We sincerely thank the reviewer for providing constructive feedback and asking thoughtful questions. We are inspired that the reviewer found our method (a) clearly written and easy to follow, (b) simple to implement, and (c) effective for image classification. We address the reviewer comments below and try to incorporate most of the suggestions.
>
> > **In-modal consistency seem to assume the distances measure in image representation space is at the same scale as that in the text representation space. Despite the input image and text space maybe inherently different, the assumption may not be true when the text does not fully describe the image (which is the majority case in web-crawled data). The reviewer further points at the counter-examples where the in-modal and cross-modal consistency might not be desirable.**
>
> We thank the reviewer for thinking profoundly about the plausible concerns with our formulation and providing useful insights into the same. We provide further clarifications that might help answer their questions: (1) As discussed in Section 2.2, the image and text representations are assumed to have unit L2-norm. (2) The reviewer accurately points out that the caption does not fully characterize the corresponding image for most web-crawled data. We concur with the reviewer's remark that raw images and captions can capture information at different resolutions. However, in an unsupervised setting, it is not apparent to what extent these distinctions should be reflected in the embedded representations. The usefulness of any information captured in a modality depends on the downstream task at hand.
>
> For example, in the reviewer's first example, let us say the eventual goal is to use the learned image and text representations to classify cats vs. dogs. In that case, even if the other objects in the image (e.g., the person) have different characteristics, this is a spurious correlation that could be ignored. We would ideally desire the in-modal distance between text and image embeddings to be similar for consistent predictions in both modalities.
>
> Similarly, in the reviewer's second example, where some captions are more complete than others, this is not a sole criterion for determining the distance between embeddings in the downstream tasks; for example, the "room" location in text-1 may not be of interest in classifying cats vs dogs.
>
> Moreover, a similar counter argument could be made against the default CLIP loss: the loss weighs each image and text pair equally, even though some text captions might be more descriptive about the images than others and therefore should be assigned a higher weight.
>
> From a practical standpoint, our evidence suggests that soft consistency regularization in the form of additional loss terms, as in Eq. 5, can be generally helpful for the downstream tasks and domain settings of interest. The relative gains can indeed be different across tasks and domains. For example, in Fig 3, while our regularizers lead to gains in fine and coarse-grained classification, we noted that the relative gains are much higher for coarse-grained settings due to the improved consistency.
>
> > **It is also not very clear whether the current advantage of CyCLIP would hold when scaling up to larger pretraining corpus. To perform a fast validation, maybe the authors can add 1M image-text pairs from SBU dataset or even 12M data from CC12M as additional pre-training data.**
>
> We thank the reviewer for asking a very pertinent question. We were aware of this limitation in the original draft and mentioned it in our Conclusion section.
>
> We would like to bring the attention of the reviewer to the Section 4.4 Figure 4(a) where we try to partially answer this question by training 4 CLIP and CyCLIP models at various pre-training dataset sizes. We find that the CyCLIP outperforms CLIP across the whole range of pre-training corpus.
>
> Additionally, we have very scarce resources for training models in the short rebuttal period, but we are trying our best to get larger scale results in the remainder of the rebuttal period. We'll update if and when we get those results.

---

> > ### Author Response · Authors · 2022-08-02
> > **Author Response: Image-Text Retrieval [2/N]**
> >
> > > **Would the added regularizers affect the performance on some applications other than image classification, for example image-to-text retrieval?**
> >
> > As requested by the reviewer, we conducted zero-shot and fine-tuned cross-modal retrieval experiments on the standard Flickr30K 1K and MSCOCO 5K test set. For both the datasets, image-to-text retrieval requires each image to retrieve one of the five relevant captions in its top K closest predictions. In contrast, the text-to-image retrieval requires each caption to retrieve the correct image (only one possible) in its top K closest predictions. This explains why we observe lower image retrieval scores over text retrieval scores. While CLIP and CyCLIP are comparable on the easier text retrieval tasks, we observe that CyCLIP outperforms CLIP across both the datasets on the Image retrieval task in both the zero-shot and fine-tune cases.
> >
> > *Abbreviation: TR - Text Retrieval, IR - Image Retrieval*
> >
> >
> > | Flickr30K (1K) | R@1 (TR) | R@5 (TR) | R@10 (TR) | R@1 (IR) | R@5 (IR) | R@10 (IR) |   |   |   |   |
> > |-------------|----------|----------|-----------|----------|----------|-----------|---|---|---|---|
> > | **Zero-shot**      |          |          |           |          |          |           |   |   |   |   |
> > | CLIP           | 88.2     | 93.9     | 95.8      | 29.9     | 57.2     | 68        |   |   |   |   |
> > | CyCLIP         | 88.1     | 93.7     | 95.9      | **30.9**     | 57.8     | **69.1**      |   |   |   |   |
> > | **Fine-tuned**      |          |          |           |          |          |           |   |   |   |   |
> > | CLIP           | 91.9     | 97       | 98        | 46.3     | 74.7     | 83.6      |   |   |   |   |
> > | CyCLIP         | 92.3     | 97       | 98.4      | **47.3**     | **76.6**    | **85.4**      |   |   |   |   |
> > |                |          |          |           |          |          |           |   |   |   |   |
> >
> > | MSCOCO (5K) | R@1 (TR) | R@5 (TR) | R@10 (TR) | R@1 (IR) | R@5 (IR) | R@10 (IR) |   |   |   |   |
> > |-------------|----------|----------|-----------|----------|----------|-----------|---|---|---|---|
> > |**Zero-shot**  |          |          |           |          |          |           |   |   |   |   |
> > | CLIP        | 82.1     | 85.6     | 87.8      | 8.4      | 19.5     | 26.6      |   |   |   |   |
> > | CyCLIP      | 82.1     | 85.6     | 87.7      | 8.6      | 20       | 27        |   |   |   |   |
> > | **Fine-tuned**   |          |          |           |          |          |           |   |   |   |   |
> > | CLIP        | 83.2     | 87.6     | 90        | 10.6     | 23.9     | 31.3      |   |   |   |   |
> > | CyCLIP      | 83.2     | 87.8     | 90.3      | **11.4**     | **25.8**     | **33.4**      |   |   |   |   |
> > |             |          |          |           |          |          |           |   |   |   |   |
> >
> > The relatively lower performance of both CLIP and CyCLIP in the zero-shot setting may be attributed to the more complicated nature of the two datasets where the models are expected to find similarities between the image and text at multiple resolutions as opposed to image classification where there is mostly single object to be matched with a simpler caption. It is not clear as to what distinctions in the raw input and text space are reflected in the embedding space too. Hence, we perform fine-tuning on both the datasets to better inform our models of the downstream datasets.
> >
> > We further create a text consistency metric that measures the proportion of the captions for which we retrieve the correct image and one of the four similar captions simultaneously over the whole dataset. This is similar to Equation 2 in our paper. We find that CyCLIP outperforms CLIP on fine tuning for both the datasets. The following table summarizes are results.
> >
> > |           | Flickr30K 1K | MSCOCO 5K |
> > |-----------|--------------|-----------|
> > | **Zero-shot** |              |           |
> > | CLIP      | 20.9         | 2.8       |
> > | CyCLIP    | **21.3**         | 2.8       |
> > | **Fine-tuned** |              |           |
> > | CLIP      | 35.4         | 3.8       |
> > | CyCLIP    | **37.7**         | **4.1**       |

---

> > > ### Author Response · Authors · 2022-08-05
> > > **(Update) Author Response: Larger Dataset Size [3/N]**
> > >
> > > > **The authors can add 1M image-text pairs from SBU dataset or even 12M data from CC12M as additional pre-training data.**
> > >
> > > As mentioned in our rebuttal, with our limited resources, we were able to complete the training of the CLIP and CyCLIP models on a larger dataset size of 4Million (CC3M + 1M from CC12M). The training hyperparameters are identical to the 3M runs.
> > >
> > > We present the zero-shot Top-1 classification accuracy results on the series of challenging datasets. We observe that CyCLIP outperforms CLIP across all the datasets showcasing better zero-shot ability as well as the robustness to natural distribution shift at a larger scale.
> > >
> > > | Top-1       | IN-1K | IN-V2 | IN-Sk | IN-A | IN-R |
> > > |-------------|-------|-------|-------|------|------|
> > > | CLIP (4M)   | 22.0  | 18.3  | 13.0  | 4.8  | 27.4 |
> > > | CyCLIP (4M) | 24.4  | 20.6  | 14.8  | 5.4  | 30.4 |
> > > | Gain (%)       | **11.1**  | **12.7**  | **13.6**  | **10.8** | **11.0** |
> > > |||||||||
> > >
> > > *Abbreviations: IN-1K - ImageNet-1K, IN-V2 - ImageNet V2, IN-Sk - ImageNet-Sketch, IN-A - ImageNet-A, IN-R - ImageNet-R*

---

> ### Comment · Reviewer_tnET · 2022-08-05
> **Response to rebuttal**
>
> Thanks the authors for the detailed response. I appreciate the authors for adding the additional experiments and discussion. The paper now looks more complete to me, and the response has cleared some of my doubts. I have raised my score and recommend accept.

---

### Author Response · Authors · 2022-08-02
**Summary of all the changes**

We sincerely thank the reviewers for their thoughtful questions and feedback. We are pleased that all tnET, KkEn, and ra5c found our paper easy to follow and well-written . We are also happy to hear that KkEn and ra5c found our contribution towards identification of inconsistency interesting and important. We are glad that KkEn found our analysis inspiring. Finally, we are pleased that ra5c found our consistency regularizers modular which can be combined with any contrastive learner. Owing to the reviewers suggestions, we have added multiple new sections (and sub-sections) in our Appendix.


1. **Added Image-text retrieval results**: In section A.7, we present our cross-modal retrieval results on the Flickr30K 1K and MSCOCO 5K test set in the zero-shot and fine-tuned settings. We observe that CyCLIP performs better than CLIP on the more challenging task of Image retrieval on both the datasets.

2. **Increased zero-shot evaluations**: As suggested by ra5c, we increase the coverage of datasets for zero-shot evaluation by 4x. The results are presented in Table 11. Additionally, we perform error-free label cleaning on the popular datasets for reliable evaluation.

3. **Additional discussion**: We provide a detailed discussion on the counter-examples presented by the tnET in Section B of the appendix.

---

### Meta-Review · Area_Chair_Ddud · 2022-08-27

**Recommendation:** Accept
**Confidence:** Certain

**Metareview:**

The authors identify the inconsistency problem in CLIP for the first time and proposes a simply but effective solution to the problem. The experiments are comprehensive and the results are convincing. Reviewers' questions are well addressed in the rebuttal. The reviewers all agreed on accepting this paper. After reading the paper, I also agree with the reviewers.

**Award:**

No

---

### Decision · Program_Chairs · 2022-09-14

Accept